# Carriers of Heterozygous Loss-of-Function ACE Mutations Are at Risk for Alzheimer’s Disease

**DOI:** 10.3390/biomedicines12010162

**Published:** 2024-01-12

**Authors:** Sergei M. Danilov, Ivan A. Adzhubei, Alexander J. Kozuch, Pavel A. Petukhov, Isolda A. Popova, Ananyo Choudhury, Dhriti Sengupta, Steven M. Dudek

**Affiliations:** 1Department of Medicine, Division of Pulmonary, Critical Care, Sleep and Allergy, University of Illinois Chicago, Chicago, IL 60612, USA; alexkozuch@gmail.com (A.J.K.); sdudek@uic.edu (S.M.D.); 2Department of Biomedical Informatics, Harvard Medical School, Boston, MA 02115, USA; iadzhubey@gmail.com; 3Department of Pharmaceutical Sciences, College of Pharmacy, University of Illinois Chicago, Chicago, IL 60612, USA; pap4@uic.edu; 4Toxicology Research Laboratory, University of Illinois Chicago, IL 60612, USA; ipopova3051@gmail.com; 5Sydney Brenner Institute for Molecular Bioscience, University of the Witwatersrand, Johannesburg 2193, South Africa; ananyo.choudhury@wits.ac.za (A.C.); dhriti.sengupta@wits.ac.za (D.S.)

**Keywords:** angiotensin I-converting enzyme, mutations, conformational changes, plasma *ACE*, screening

## Abstract

We hypothesized that subjects with heterozygous loss-of-function (LoF) *ACE* mutations are at risk for Alzheimer’s disease because amyloid Aβ42, a primary component of the protein aggregates that accumulate in the brains of AD patients, is cleaved by ACE (angiotensin I-converting enzyme). Thus, decreased ACE activity in the brain, either due to genetic mutation or the effects of ACE inhibitors, could be a risk factor for AD. To explore this hypothesis in the current study, existing SNP databases were analyzed for LoF *ACE* mutations using four predicting tools, including PolyPhen-2, and compared with the topology of known *ACE* mutations already associated with AD. The combined frequency of >400 of these LoF-damaging *ACE* mutations in the general population is quite significant—up to 5%—comparable to the frequency of AD in the population > 70 y.o., which indicates that the contribution of low ACE in the development of AD could be under appreciated. Our analysis suggests several mechanisms by which ACE mutations may be associated with Alzheimer’s disease. Systematic analysis of blood ACE levels in patients with all *ACE* mutations is likely to have clinical significance because available sequencing data will help detect persons with increased risk of late-onset Alzheimer’s disease. Patients with transport-deficient *ACE* mutations (about 20% of damaging ACE mutations) may benefit from preventive or therapeutic treatment with a combination of chemical and pharmacological (e.g., centrally acting ACE inhibitors) chaperones and proteosome inhibitors to restore impaired surface ACE expression, as was shown previously by our group for another transport-deficient ACE mutation-Q1069R.

## 1. Introduction

Alzheimer’s disease (AD) is the most common cause of dementia, afflicting~35 million people worldwide. Extracellular β-amyloid protein (Aβ) deposition in senile plaques [1] is a primary pathologic manifestation of AD. Several genes have been linked to AD risk, but AD etiology remains incompletely understood. Familial AD is associated with mutations in amyloid precursor protein *APP*, and presenilins (*PS1* and *PS2*), while late-onset AD is associated with mutations in *APOE* and other genes involved in vascular dysfunction, immunity and inflammation, cholesterol metabolism, endocytosis, and ubiquitination [2].

The primary fragments generated from amyloid precursor protein (APP) are Aβ42 and Aβ40. Aβ42 is an important early contributor to amyloid deposition during AD development. In contrast, Aβ40 has antioxidant properties and can inhibit Aβ42 deposition in brain tissue. A decrease in Aβ40 is, therefore, a possible step in AD [3]. Angiotensin-converting enzyme (ACE) is able to convert Aβ42 to Aβ40 in the brain by the N-domain active center of ACE [4]. Therefore, a decrease in tissue ACE activity (either due to constitutive intrinsic mutations or ACE inhibitor treatment) could be associated with Alzheimer’s disease (AD) pathogenesis. An initial hypothesis about the possible association of ACE and Alzheimer’s risk [5] was initiated by observation that a majority of centenarians have a variant of the *ACE* gene (DD genotype) that is associated with the highest level of ACE [6]. Therefore, it was hypothesized that low levels of tissue ACE or ACE inhibition may confer increased risk of AD. Numerous papers published by the Kehoe’s [7], Zou’s [4], and Muller’s [8] groups and many others were generally supportive of this hypothesis. 

Angiotensin I-converting enzyme (ACE, CD143) is a Zn^2+^ carboxydipeptidase that metabolizes a number of metabolically active peptides and thus plays key roles in the regulation of blood pressure, in the development of vascular pathology, and in innate immunity. ACE is expressed on the surface of endothelial cells, absorptive epithelial and neuroepithelial cells, immune cells (macrophages, dendritic cells), and some neurons. The level of ACE expression in individuals is strongly genetically determined (e.g., the ACE I/D polymorphism), but it is also influenced by glucocorticoid and thyroid hormones. Blood ACE normally originates by proteolytic cleavage from endothelial cells, primarily lung capillary endothelium, and generally reflects the level of tissue ACE expression in a given individual. In healthy individuals, blood ACE levels are very stable throughout their life span, whereas in granulomatous diseases (e.g., sarcoidosis) and Gaucher’s disease, blood ACE activity is significantly increased (reviewed in [9,10]).

We have developed a novel approach for characterization of ACE status in the blood—ACE phenotyping—and established that normal values of blood ACE levels are 50–150% of control pooled plasma [11]. Therefore, measurement of ACE activity/levels in the blood, after adjustment for hormonal status and ACE genotype, may help to identify individuals with the lowest levels of tissue ACE expression who may be at risk for the development of late-onset Alzheimer’s disease.

We have analyzed an existing database for ACE mutations and have identified that ~4–5% of the general population may have damaging, loss-of-function (LoF) *ACE* mutations that may result in very low tissue ACE expression and, therefore, increased risk for the development of late-onset Alzheimer’s disease. We present here an example of such an analysis and also describe different mechanisms by which these ACE mutations may initiate Alzheimer’s disease. The most important (and clinically relevant) conclusion of this analysis is that patients with transport-deficient ACE mutations (about 20% of all damaging ACE mutations) may benefit from preventive or therapeutic treatment with a combination of chemical and pharmacological (e.g., centrally acting ACE inhibitors) chaperones and proteosome inhibitors to restore impaired surface ACE expression. This approach has previously been reported for another transport-deficient *ACE* mutation (Q1069R), which was causal for renal tubular dysgenesis-RTD [12]. These findings may assist the urgent need for the development of novel drug targets for AD treatment as well as identify new biomarkers for early evaluation of disease risk.

## 2. Methods

### 2.1. Analysis of the Existing Databases for ACE Mutations (PolyPhen-2 and Other Engines)

Variation data and population allele frequencies for the ACE gene (ENST00000290866, GRCh38/hg38 genome assembly) were extracted from the dbSNP database [13] (https://www.ncbi.nlm.nih.gov/snp/ accessed on 12 September 2022) using the UCSC Table Browser tool. Variation data were downloaded from the UCSC Genome Browser (build 155) on 12 September 2022. Variants with a reported anomaly or problem (e.g., mapping errors) in the variant annotation, as well as variants from the datasets with total number of chromosomes sampled below 100, were excluded. Variants retained after filtering were mapped to the ACE protein sequence (UniProt accession: P12821) and functional predictions for all mapped nsSNVs were fetched from the dbNSFP v4.3a database [14]. Predictions and scores from four different in silico tools which account for evolutionary conservation and structural features were utilized—PolyPhen-2 [15], SIFT4G [16], VEST4 [17], and REVEL [18]. Majority vote predictions (benign/damaging) were calculated for cases where more than two of the tools independently delivered the same prediction and known disease association annotations were obtained from the ClinVar [19] database released 22 January 2022 (Appendix A).

### 2.2. Localization of AD Associated ACE Mutations on the N- and C-Domains of ACE

For Figure 1, Figure 2 and Figure 3, coordinates of X-ray model of the N-domain of human ACE (PDB: 3NXQ [20] and C-domain PDB: 2XY9 [21]) were downloaded from the PDB. The hydrogen atoms were added, and the resulting models were rendered in PYMOL. For Figure 4, the structure of the transmembrane domain was obtained using the homology module in the Molecular Operating Environment (MOE www.chemcomp.com).

The transmembrane sequence R1227-R1250 was modeled as a helix. The cytoplasmic domain R1250-S1277 sequence was prepared in MOE, subjected to the “structure preparation” procedure and hydrogen atoms were added using the Protonate 3D algorithm. The resulting cytoplasmic domain was minimized utilizing AMBER14:EHT forcefield in MOE [22] until the RMS gradient reached 0.01 kcal/mol/Å^2^, solvated in a periodic box with water and 0.1 M KCl using the MOE “Solvate” module. The water box was extended at least 10 Å from the protein. The energy of the resulting structure was minimized with periodic boundary conditions (PBC) enabled and utilizing AMBER14:EHT forcefield in MOE [23] until the RMS gradient reached 0.01 kcal/mol/Å^2^. The “Molecular Dynamics” module in MOE was used to prepare the resulting structure for MD simulations using NAMD software, version 14, Linux-x86_64-multicore-CUDA [23]. Each solvated complex was heated to 300 K for 10 ns and equilibrated in a series of steps: 10 ns in the NVT ensemble at 300 K, 20 ns in the NPT ensemble at 1 atm and 300 K, and finally for 1 microsecond in the NPT ensemble at 1 atm and at 300 K. The non-bonded interactions were switched at 8 Å and zero smoothly at 10 Å (cutoff 10 Å, switchdist 8 Å, nonbondedScaling 1, pairlistdist 11.5 Å, limitdist 0.5 Å). The temperature was maintained using of Langevin dynamics with a damping coefficient of 5.0 ps^−1^. In PBC, the wrapAll parameter was used to calculate all the coordinates around periodic boundaries. Electrostatic interactions in PBC were treated using the Particle Mesh Ewald (PME) method and PMEGridSpacing set at 1.0 Å. Covalent bonds with hydrogen atoms were kept rigid using ShakeH with the following parameters rigidbonds—all, rigidtolerance—10^−6^ Å, and water molecules were kept rigid using the Settle algorithm. The time step size for the integration of each step of the simulation was 2 fs. All the other parameters were unchanged from the default settings in the NAMD software. 

The resulting structure of the cytoplasmic domain was stripped of the buffer molecules and attached to the helical transmembrane domain generated previously. The resulting structure D1222-S1277 was prepared for MD simulation in QwikMD [24] in VMD, version 1.9.4alpha55 Linux (http://www.ks.uiuc.edu/Research/vmd/) [25]. Specifically, the ACE protein was placed in a 100 by 100 angstrom bilayer of 3-palmitoyl-2-oleoyl-D-glycero-1-phosphatidylcholine (POPC) membrane and solvated with 150 mM NaCl water buffer spanning 15 angstrom at the top and the bottom of the membrane and the protein. The CHARMM36 force field [26] and the TIP3 [27] water model were used to describe the system as specified in QwikMD. The resulting system was minimized for 2000 steps, annealed for 140,000 steps, and equilibrated for 500,000 steps using the default harmonic constraints on the backbone atoms of the protein in the NpT ensemble at 300 K and 2 fs time step. All the other parameters were unchanged from the default settings in the QwikMD module. The harmonic constraints were removed, and the system was subjected to equilibration for 100 ns in the NpT ensemble followed by a 260 ns production run. 

The R1250Q and R1257S mutants were generated using QwikMD “Structure Manipulation” module and subjected to the same MD simulations protocol. The helical part of the protein was detected using the Bendix module in VMD [28]. The angle between the ACE helix and the lipids was calculated using the coordinates of CA atoms in Q1224 and L1247 in WT ACE, R1227 and Q1249 in the R1250Q mutant, and R1227 and L1247 in the R1257S mutant. 

The distances between Q1225 and the charged residues in the cytoplasmic domain, R1250, R1255, R1257, R1261, E1271, E1273, and R1275, were determined between the average position of the corresponding residues using distance.tcl script in VMD. The distances for all the pairs of the residues were grouped into 10 bins individually for each pair.

### 2.3. Chemicals

ACE substrates, Z-Phe-His-Leu (ZPHL)—Cat.# 4000599 and Hip-His-Leu (HHL) Cat.# H1635 were purchased from Bachem Bioscience Inc. (King of Prussia, PA, USA) and Sigma (St. Louis, MO, USA), respectfully. Other reagents (unless otherwise indicated) were obtained from Sigma (St. Louis, MO, USA). 

### 2.4. Antibodies

Antibodies used in this study included a set of 26 (mAbs) to human ACE, recognizing the native conformation of the N- and C-domains of human ACE [29,30].

### 2.5. Human Tissues

Lung, kidney, and brain (frontal cortex) (as fresh–frozen tissues from one donor) were purchased from ProteoGenex, (Inglewood, CA, USA). Homogenates from these tissues were prepared in PBS (1:9 weight/volume) using tissue homogenizer and then Triton X-100 were added to final concentration 0.25% to solubilize membrane-bound ACE. Supernatants of such homogenates were used as a source of lung, kidney, and brain ACEs in dilutions 1/20, 1/40 and 1/5, respectively.

### 2.6. ACE Activity Assay

ACE activity in the homogenates of human tissues was measured using a fluorimetric assay with two ACE substrates, 2 mM Z-Phe-His-Leu or 5 mM Hip-His-Leu [31]. Calculation of ZPHL/HHL ratio [32] was performed by dividing fluorescence of the reaction product produced by ACE sample with ZPHL as a substrate to that with HHL. 

### 2.7. Immunological Characterization of the Brain ACE

Microtiter (96-well) plates (Corning, Corning, NY, USA) were coated with anti-ACE mAbs via goat anti-mouse IgG (Cat.# 31170, ThermoFisher-Invitrogen, Carlsbad, CA, USA) bridge, and incubated with homogenates of human tissues or with plasma/serum samples. After washing of the unbound ACE, plate-bound ACE activity was measured by adding ACE substrate (Z-Phe-His Leu) directly to the wells. The level of ACE immunoreactive protein, using strong mAb 9B9, was quantified as described previously [31]. Conformational fingerprinting of ACE using a set of mAbs to different epitopes of ACE was performed as described [29,30].

### 2.8. Allele Frequencies and Age of ACE Variants

Allele frequencies of ACE variants in different global populations were retrieved from the 1000 Genomes Project, gnomMAD and [33]. The presence of these alleles in Neanderthal genomes was interrogated using https://neandertal.ensemblgenomes.org/index.html. Age estimation of selected variants was performed using https://human.genome.dating/.

### 2.9. Statistical Analysis

Values of ACE activity precipitation with different mAbs characterizing ACE conformation were means ± SD from at least 2 independent experiments with du- or triplicates in each experiment. Significance was analyzed using the Mann–Whitney test.

## 3. Results and Discussion

### 3.1. Aβ42 Hydrolysis by ACE Occurs In Vivo in Humans and Plays a Role in AD Development

The observations that the D allele of the ACE gene I/D polymorphism is associated with increased lifespan [6] inspired the hypothesis and subsequent confirmation [5] that the D allele (associated with increased levels of ACE expression [34]) may be protective for the development of AD, while the I allele (associated with decreased ACE expression) may result in increased risk. A second series of mechanistic observations supporting the hypothesis that ACE may be associated with AD is that ACE, and particularly the N-domain active center, degrade in vitro amyloid beta peptide (Aβ42), which is an essential component for amyloid deposition [4].

The absence of an observed effect on elimination of Aβ42 in *ACE* KO mice [35] resulted in the field prematurely and erroneously concluding that ACE does not play a role in AD development via modulation of Aβ42 hydrolysis. There are several possible explanations for this apparent discrepancy in the data evaluating the role of ACE in AD pathophysiology. These include the major differences in ACE genotype effects in mice and humans. *ACE* KO mice are viable (albeit with decreased male fertility and impaired kidney function [36]), while the vast majority of humans genetically deficient in ACE die in utero due to RTD [37]. This much more severe genotype of *ACE* KO in humans compared to mice is likely due to differences in nephrogenesis [38]. In addition, human monocytes increase their surface expression of ACE by 50–150-fold during differentiation into macrophages or dendritic cells [39], while mouse dendritic cells are characterized by much lower ACE surface expression–less than in peripheral monocytes [40].

Another study supports the hypothesis that ACE-dependent hydrolysis of amyloid peptide Aβ42 occurs in vivo. C-domain-dependent ACE activity in the serum of control and AD patients was similar, while serum N-domain-dependent Aβ42 to Aβ40 conversion was lower in AD than that in controls [3]. Studies utilizing various mouse models of AD confirmed that ACE can degrade Aβ42 in vivo (even in mice), and that low ACE activity in the brain (disease–associated or ACE inhibitor-induced) enhanced the level of amyloid plaque [3]. 

### 3.2. Analysis of ACE Mutations Decreasing Blood ACE Levels

Recently we analyzed an existing database of ACE mutations that **increased** blood ACE levels [41]. A prior study describing the association of ACE mutation R1250Q (mature ACE numbering—rs4980) with Alzheimer’s diseases [42] inspired us to analyze the dbSNP database [13] for *ACE* mutations that result in **decreased** blood ACE levels. We identified in this database ~500 *ACE* missense mutations predicted as damaging, according to PolyPhen-2, SIFT4G, VEST4, and REVEL software mutation effect prediction tools (Appendix A and Table 1).

Using only concordant predictions from all four tools, about 400 of those missense variants in the *ACE* gene were found to be potentially damaging. Summing up minor allele frequencies (MAFs) for the *ACE* missense variants reported in gnomAD exome dataset (see Appendix A), we obtained the estimated abundance of potentially **damaging** risk alleles segregating in the general population at 0.81%—Table 1. Note, common alleles (AF ≥ 0.5%) were excluded from this calculation to avoid bias due to strong genetic linkage between such alleles located within the same gene.

Despite slightly lower cumulative frequencies (MAF) for the variants predicted as damaging by all four tools, we did not observe significant differences in the means of the cumulative MAFs between damaging vs. benign allele frequency distributions (Table 1, Kolmogorov–Smirnov one-sided test *p*-value < 0.134). These observations, along with the significant number of putatively damaging rare missense variants found in the *ACE* gene (Table 1, Appendix A), indicate the absence of strong genetic selection against damaging ACE mutations during evolution. This absence is possibly due to the small fitness effect size of individual autosomal heterozygous alleles, the presence of common protective alleles in other genes, and/or relaxation of selection in humans due to rapid population size growth, among other factors [43]. However, statistically significant associations of less rare ACE mutations with AD development strongly support a causal role for low ACE levels in AD as demonstrated in numerous publications [5,44], including GWAS–based analysis [45].

The high predicted frequency of damaging *ACE* mutations suggests that a significant number of individuals will have very low ACE activity. These data inspired the hypothesis that carriers of **heterozygous** LoF *ACE* mutations may be at risk for Alzheimer’s because they have only one functional allele of ACE and thus half the normal level of ACE activity. This, logically, will result in more Aβ42 accumulation; most **homozygous** carriers of LoF (damaging) *ACE* mutations will die in utero [37,38].

Fifty-one of these predicted LoF mutations occur in exon 1, which encodes the conserved signal peptide (Group I in Appendix A) rich in leucine repeats. Therefore, most of the mutations observed in exon 1 of the *ACE* gene should disrupt this signal sequence and thus prevent the normal translocation of ACE protein to the cell surface [37]. In total, 6 (out of 51) of these *ACE* mutations were identified in patients with RTD [37], demonstrating that the carriers of these *ACE* mutations (homozygous or compound heterozygous) have negligible ACE expression. Patients that are heterozygous for these mutations should have half the normal level ACE expression. Therefore, there is indirect evidence for a dramatic decrease in ACE expression (and blood ACE levels) in carriers of at least six of these LoF mutations in the signal peptide region of ACE (Appendix A lists blood ACE levels and mutations with low blood ACE). The same scenario exists with frameshift mutations (due to indels) or mutations introducing stop codons (Group II in Appendix A). Of 131 such *ACE* mutations (which are by definition damaging) identified in databases, our analysis indicated that 28 mutations are associated with RTD. These data further support the conclusion that heterozygous carriers of such *ACE* mutations will have half the normal level of ACE expression, which we predict will increase risk for AD development. Further confirmation of the hypothesis that carriers of heterozygous LoF *ACE* mutations are at risk for AD comes from a recent study, in which six patients with heterozygous LoF *ACE* mutations were diagnosed with late-onset AD with evidence of abnormal Aβ deposition [46]. 

Our analysis identified another 390 missense *ACE* mutations (Appendix A) which can be classified as “probably damaging” (according to PolyPhen-2). Previously, we have reported two studies that validate predictions made by PolyPhen-2 that correctly classified “probably damaging” ACE mutations. First, we recently characterized ACE in a cohort of 300 unrelated patients that contained two patients with very low ACE levels (45% of mean value) due to a p.Tyr244Cys mutation of *ACE*—rs373025 (Y215C-mature ACE numbering [11]). This mutation is listed in Appendix A (and Table 2) and has a very high MAF (more than 1% in general population). 

Secondly, we previously found that heterozygous patients with another “probably damaging” *ACE* mutation (Q1069R) also had very low blood ACE levels that average ~36% of the mean in the general population [12]. There is additional indirect evidence that these predictions by PolyPhen-2 scores are correct. Carriers of six such “probably damaging” missense LoF *ACE* mutations (as homozygotes or compound heterozygotes) developed RTD and had low ACE expression, and thus blood ACE activity (Appendix A), indicating that these six mutations characterized as “probably damaging” truly reduce ACE function.

The combined frequency of all “probably damaging” *ACE* mutations as predicted by PolyPhen-2 scores was 3.5% in general population (Appendix A). The addition of all carriers of “possibly damaging“ *ACE* mutations was another 1.8%. Moreover, a carrier with one *ACE* mutation considered as benign by PolyPhen-2 score—see R1180P (rs5381166970) in Appendix A, also developed RTD [50] (i.e., non-functional allele of *ACE* gene), suggesting that the combined frequency of LoF *ACE* mutations that may participate in AD development could be as high as 5% in the general population. Such a high percentage is comparable with the frequency of AD in the population > 70 y.o., which indicate that contribution of low ACE in the development of AD could be underappreciated. 

Therefore, an important next step will be the measurement of blood ACE activity/levels in the carriers of “possibly damaging “ACE mutations as well as in patients with “benign” *ACE* mutations, in order to identify all *ACE* mutations that result in significantly decreased blood ACE levels (i.e., significantly decreased tissue ACE) that may participate in the development of late-onset AD. We already initiated such a study, using blood from subjects with ACE mutations that were found in a sequencing database of more than 10,000 exomes [51].

Furthermore, it is reasonable to hypothesize that patients with any LoF *ACE* mutations and the II genotype (or with polymorphic variants in the promoter -rs4291 or rs1800764—that also result in decreased ACE expression [52]) will be at even higher risk for Alzheimer’s disease because the levels of ACE expression will be even lower in these individuals.

Several *ACE* mutations, already associated with AD, are listed in Table 2. The combined frequency of these *ACE* mutations is approximately up to 3 % in the general population. If the frequency of other *ACE* damaging mutations are added, then the combined frequency would be higher than the frequency of Alzheimer’s disease in the 70+ y.o. population. This suggests that other modifier genes or protective effectors are present that may help to compensate for the harmful effect of the “damaging” ACE mutations and low ACE levels. 

### 3.3. Global Distribution and Age of ACE Variants

Next, we compared the geographic distribution and allele ages of the most frequent seven ACE variants that are associated with Alzheimer’s diseases (in the Table 2) based on the 1000 Genomes Project (1KGP), African Genome Variation database (AGVP), and the gnomAD datasets. The results are summarized in Appendix A and indicate that four are globally rare variants occurring at ~1% or lower frequency across continents in both 1KGP and gnomAD datasets. Two of these, T887M (rs3730043) and R1250Q (rs4980), had similar frequencies in Africa and in Europe in the 1KGP dataset. N1007K (rs142947404) was found only in two of the 1KGP populations, Italian (TSI) and Peruvian (PEL), leading to the speculation of a possible link between the distribution of the variant to South European gene flow to South Americas during the last millennia. However, due to the rarity of this variant, much larger and targeted datasets are required for further characterization. Variant E738K (rs148995315) was absent in 1KGP and was extremely rare and restricted only to European ancestry populations in the much larger gnomAD dataset [53].

The other three SNPs demonstrated a trend towards being African-specific, having several fold higher frequencies in African populations compared to all other continents. The allele ages of these variants (Appendix A), estimated using the Human Genome Dating Atlas, suggest that at least two of these evolved prior to the out-of-Africa migration but remained on the continent until recently. The most dramatic of these is the R1257S *ACE* mutation (rs4364), which occurs in approximately 13% in those of African descent but is absent or extremely rare in populations from the other continents. This variant exhibited almost two-fold higher frequency in West African Niger–Congo speakers (MSL and GWD) compared to Niger–Congo speakers form Central-West (YRI and ESN) and East Africa (LWK and Baganda) (Appendix A). Moreover, the much lower frequency (about 2%) of these variants in the non-Niger–Congo speaking Ethiopian population further highlights the extreme intra-continental variation of this SNP. 

SNP D563G (rs12709426) demonstrated a similar trend (shown in Appendix A) of higher frequency in West African populations, although the differences in allele frequencies within the continent was not as extreme. In contrast, A232S (rs4303) exhibited frequency differences within each of the African geographic regions rather than between them.

Recent whole genome studies, especially those based on African populations, have shown that the survey of the global distribution of variants may identify curated disease-related variants (such as those in the ACMG list) in need of revision [54]. Given the commonality of the three ACE variants described above in Niger–Congo speakers and the absence of known elevations in AD in Africa, specifically in West Africa, suggests that associations of these three SNPs to AD is highly questionable. However, it remains possible that these variants may confer increased risk in conjunction with other variants such as the Epsilon allele of APOE [43]. Moreover, their extreme differentiation within the continent suggests that at least some of these alleles may be associated to one or more key biological functions. Further investigation is needed to evaluate these possibilities.

We also attempted to trace these possible AD-associated *ACE* mutations (Table 2) to the published Neanderthal genomes (https://neandertal.ensemblgenomes.org/index.html), focusing on the four European-related variants (rs3730043, rs4980, rs142947404, rs148995315). None of these variants were found in the few Neanderthal genomes available. Given the pattern of distribution within Europe and the extremely low frequency of variants, much larger datasets of Neanderthal, ancient, and current Europeans genomes would be required for investigating the direction of flow of these alleles within the continent.

### 3.4. Localization of AD-Associated ACE Mutations and Possible Mechanism of Its Action

Blood ACE levels associated with these *ACE* mutations are listed in Table 2 and Appendix A. These data combined with previously published literature suggest several possible mechanisms for the association of these *ACE* mutations with AD.

(1) ***Low levels of ACE expression***: This possible mechanism is the most simple and straightforward. Low ACE expression (and activity) occurs because of one non-functional allele in the *ACE* gene, or due to indels or stop codons (Group II in Appendix A with 131 mutations). A convincing example of this mechanism is provided by six such mutations described above [46]. However, the combined frequency of such ACE mutations is not high—less than 0.2% in the general population. Gene therapy (CRISPR-based) may theoretically help such patients but is not currently available. 

(2) ***Low levels of ACE activity***: This potential mechanism occurs when mutations occur in the active center residues discussed in [55,56,57]. However, only such mutations that occur in the N-domain active center are likely to be associated with AD by this mechanism. Only the N-domain active center hydrolyzes Aβ42, which is an important step for amyloid deposition [4].

(3) ***Low levels of surface ACE protein expression***: Another potential mechanism involves mutations that result in decreased surface ACE expression (similar to what we described previously for the Q1069R mutation [12]). Examples include mutations in the C-domain, such as E738K, T887M, and N1007K (mature somatic ACE numbering listed in Table 2 and shown on Figure 1), as well as many others shown in Group III (missense mutations) of Appendix A. These mutations could contribute to AD through decreased ACE function.

**Figure 1 biomedicines-12-00162-f001:**
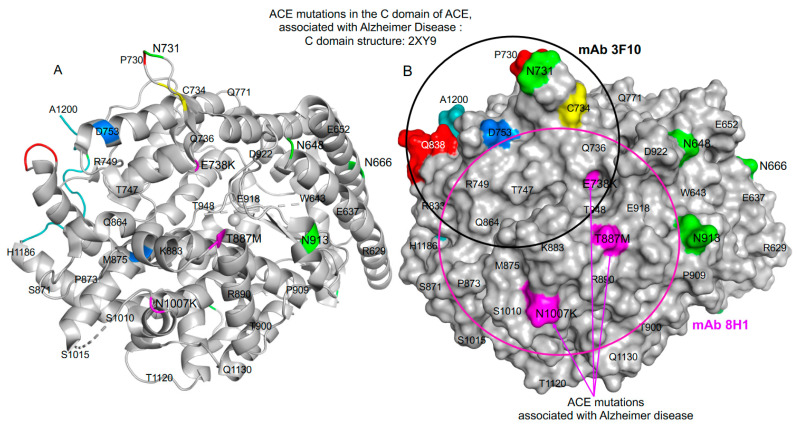
Localization of AD-associated ACE mutations on the C-domain of ACE. Shown is the crystal structure of the C-domain fragment of human ACE in which the 36 amino acid residues unique to tACE were deleted (PDB 2XY9) using ribbon (**A**) and molecular surface (**B**) representations. Key amino acids are denoted using somatic ACE numbering. The ribbon and surface are colored gray, with specific amino acid residues colored as following: asparagines of the putative glycosylation sites are highlighted in green; AD-associated ACE mutations are highlighted in magenta; the C-terminal end of this C-domain fragment (1187–1200) is marked in light blue; amino acid residues (837AQH) that are crucial for mAb 1B3 (directed to the C terminal end) and P730 (crucial for mAb 3F10) are marked in red; the end of the cysteine bridge (C728–C734) at C734 is marked in yellow. The epitope for mAb 8H1 to the C-domain (that spans three ACE mutations)—magenta circle with a diameter 30 Å, which corresponds to approximately 700 Å^2^ of the area covered by this mAb. The epitope for mAb 3F10 (that spans one AD-associated ACE mutation in the C-domain of ACE (E738K)) is shown as a black circle.

Other examples of such transport-deficient ACE mutations in the C-domain are R828H (rs146089353) and R1180P (rs5381166970), which are confirmed to cause low ACE surface expression [37,50]. Some of these mutations may be detected using mAbs directed to the epitopes involving these mutations (e.g., T887M using mAb 8H1 as has been previously reported in Figure 4 by us [30]), potentially providing a clinically useful diagnostic test.

It is more complicated to determine the possible mechanisms for AD-associated *ACE* mutations localized in the N-domain. We previously identified an *ACE* mutation in the N-domain that results in lower blood ACE activity and is likely transport-deficient: rs373025 (Y215C) [11]. This mutation occurs with fairly high frequency—1068 per 100,000 (Table 2 and Appendix A) and has been associated with Alzheimer’s disease [44,45,47]. The predicted negative effect of the Y215C substitution on ACE expression (Appendix A), including PolyPhen-2 (Appendix A), is supported by the significant decrease in blood ACE levels we observed when ACE phenotyping the carriers of this mutation was performed [11]. These observations in combination with the abovementioned studies prompted us to hypothesize that, in addition to Alzheimer’s disease development, genetically determined low ACE expression may be a contributing factor for systemic sclerosis/scleroderma pathophysiology in some patients. Deficiency in ACE due to ACE inhibition, genetic manipulation, and/or mutations also causes other severe disease phenotypes, including defects in fetal development, hypotension, inability to concentrate urine, structural renal defects, anemia, and reduced male fertility (reviewed in [10]). Therefore, it is logical to hypothesize that some individuals with low (borderline) values of ACE expression in utero due to this Y215C mutation, may develop kidney abnormalities during embryonic development (similar to the effects of taking ACE inhibitors during pregnancy [58]). As a result, patients with such *ACE* mutations may have a decreased rate of survival into their 70s, when late onset Alzheimer’s disease is most likely to become clinically detectable.

R230H is another example of a transport-deficient *ACE* mutation in the N-domain (rs370903033, listed in Appendix A) that is associated with low ACE activity [37] as is likely A232S (rs4303, listed in Table 2 with localization shown on Figure 2).

**Figure 2 biomedicines-12-00162-f002:**
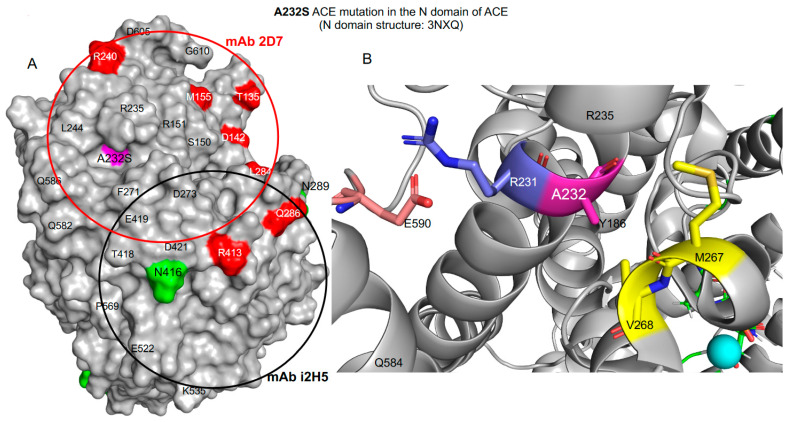
Localization of the A232S mutation on the N-domain of ACE. Shown is the crystal structure of the N-domain fragment of human ACE, where 7 amino acid residues determining putative glycosylation sites were mutated (PDB 3NXQ), using molecular surface (**A**) and ribbon (**B**) representations. Key amino acids referred to in the text are denoted using somatic ACE numbering and colored as follows: asparagines of the putative glycosylation sites are highlighted in green, and AD-associated ACE mutations are highlighted in magenta. The epitopes for mAbs 2D7 (red) and i2H5 (black) on the N-domain are shown using circles with a diameter of 30Å, which corresponds to an area of approximately 700Å^2^ covered by these mAbs.

Based upon comparison of the hinge regions found in “open” and “closed” forms of ACE (described in [59]), the A232S mutation is unlikely to have a pronounced effect on ACE enzymatic machinery or binding of ACE substrates. This mutation is on the surface in the region that does not move during opening and closing. The effect on substrate binding is likely to be minimal because A232 is relatively distant from the binding site for regular (short) peptides. Binding of longer substrates (such as Aβ42) may be affected, but the effect is expected to be minimal due to the small difference in bulk between alanine and serine sidechains. The A232S mutation also is unlikely to affect the important salt bridge between R231 and E590 [50]. The mutation is on an adjacent side of the helix that includes R231, points aways from R231, and is too small to cause substantial interference with the salt bridge between R231 and E590. A232 is located in a hydrophobic cavity formed by M267 and V268. Hypothetically, this A232S mutation could affect the strength of hydrophobic interactions at this site by reducing the hydrophobic contact and forming a new hydrogen bond with M267 (C=O) in an adjacent short helix. How this effect may alter the protein/epitope shape/interactions is difficult to predict without additional extensive computational simulations (and blood samples from carriers of this mutation are not available for our analysis).

Additional ACE mutations that cause transport deficiency are likely to be identified among 1047 missense ACE mutations listed in Appendix A. GWAS analysis, which was instrumental in finding AD-associated ACE mutations (listed in Table 2), can identify only correlations between common variants with a lower impact on risk for AD and cannot determine rare coding variants with high pathogenicity [45]. Fifty-one Group I mutations in the signal peptide (Appendix A) also relate to transport deficiency. We analyzed blood ACE levels (described in Appendix A) from a study quantifying 4907 plasma proteins (including ACE) in 35,559 Icelanders using SomaScan technology [48]. There were 22 missense *ACE* mutations found in this Iceland cohort, including at least 3 that were associated with substantial **decreases** in blood ACE levels (Appendix A). It is likely that several *ACE* mutations from this list will be transport-deficient, including mutation Y215C (rs37300025) [11]), which occurs in more than 1% of the general population.

The combined frequency of all transport-deficient ACE mutations may be significant (more than 1.5% in the general population), which theoretically may be treated using a combination of chemical (sodium butyrate) and pharmacological (central-acting ACE inhibitors) chaperones and proteosome inhibitors to restore surface expression of the mutant ACE, as we demonstrated previously with another transport-deficient ACE mutation Q1069R [12]. We speculate that a positive effect on cognitive functions of central acting ACE inhibitors in some patients with AD [60] may be explained by restoration of the impaired transport of the unrecognized mutant ACEs to the cell surface by ACE inhibitors, which are known to be effective pharmacological chaperones for transport-deficient *ACE* mutants [12].

(4) ***Impaired Aβ42 cleavage***: In addition to transport-deficient *ACE* mutations that may be detected by lower blood ACE activity (e.g., Y215C [11]), other *ACE* mutations in the N-domain could demonstrate normal ACE activity with conventional (short) substrates but have compromised binding and hydrolysis of longer substrates, such as Aβ42. An example of such divergence between Aβ42/Aβ40 converting activity and angiotensin-converting activity was demonstrated just recently for cells expressing ACE and presenilin 1 mutations. PS1 deficiency (quite unexpectedly) impaired both Aβ42/Aβ40 converting activity and angiotensin-converting activity. Some PS1 mutants restored angiotensin-converting activity, but were not able to restore Aβ42/Aβ40 converting activity [61].

It is intriguing to hypothesize that patients with the D563G mutation (and possibly others in the N-domain) may be associated with AD because this mutation is likely to specifically decrease Aβ42 cleavage (or binding) by the N-domain active center due to significant conformational changes. However, this mutation may not prevent cleavage of conventional substrates by the C-domain active center. There are two relevant observations. First, D563 is located on the side opposite to the “lid” opening of the protein, which is where ACE substrates enter the binding site (Figure 3). Second, there are several acidic and basic residues in proximity to D563 that form an extensive network of electrostatic and hydrogen bond interactions. Although D563 is excluded as part of any hinge region [62], most likely because it does not move, D563 still may play a key mechanistic role by serving as a rigid spring that holds the bottom and the top portions of the ACE “jaws” and provides the necessary plasticity to allow binding of the substrates and their release after hydrolysis is complete. The D563G mutation leads to a loss of multiple interactions and conformational rigidity because glycine does not have a sidechain, likely resulting in a loss of stiffness of the “spring” and disruption of the catalytic machinery. 

**Figure 3 biomedicines-12-00162-f003:**
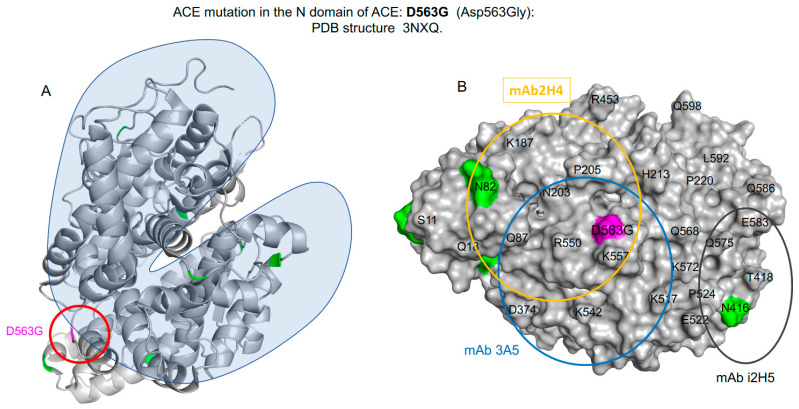
Extended view of AD-associated mutation D563G on the N-domain of ACE. (**A**) Effect of the D563G *ACE* mutation on the hinge-bending movement on the N-domain of ACE. (**B**) The crystal structure of the N-domain fragment of human ACE (PDB 3NXQ) is shown as molecular surface presentation (as in Figure 2A). The epitopes for mAbs 3A5 on the N-domain are shown using circles with a diameter of 30Å, which corresponds to an area of approximately 700Å^2^ covered by these mAbs.

Figure 3B demonstrates that the D563G mutation is located in the center of the epitope for the potent anti-catalytic antibody 3A5 [63,64]. In the contrast to classical anti-catalytic mAbs which bind to the closed entrance ofthe N-domain active center for substrates [64], the anti-catalytic effect of mAb 3A5 is based on conformational changes in the whole N-domain. After binding of this mAb to the N-domain, no other mAbs could interact with the N-domain [63]. Binding by mAb 3A5 in the region of D563 likely disrupts the dynamics of “lid” opening and closing that leads to a conformation of ACE that is not recognized by other antibodies. Because these mAbs [63] were developed for the conformation of ACE where the D563 “spring” functions normally, they are unable to recognize the new conformation of ACE caused by the binding of mAb 3A5 to the D563 area. Based on this information, we also can predict that mAb 2H4 changes binding to this mutant (D563G) (Figure 3B; see also Appendix A in [30]).

(5) ***ACE mutations in the cytoplasmic tail***: Patients with R1250Q and R1257S mutations localized in the cytoplasmic tail of ACE could be associated with AD pathophysiology by another mechanism that may include impaired phosphorylation signaling [65]. To understand the effect of these mutations on ACE interactions with the membrane, we performed MD simulations of the transmembrane and cytoplasmic domains of ACE (Asp1222–Ser1277) in 1-palmitoyl-2-oleoyl-sn-glycero-3-phosphocholine (POPC) lipid membrane for WT ACE and both R1250Q and R1257S mutants (Figure 4). 

**Figure 4 biomedicines-12-00162-f004:**
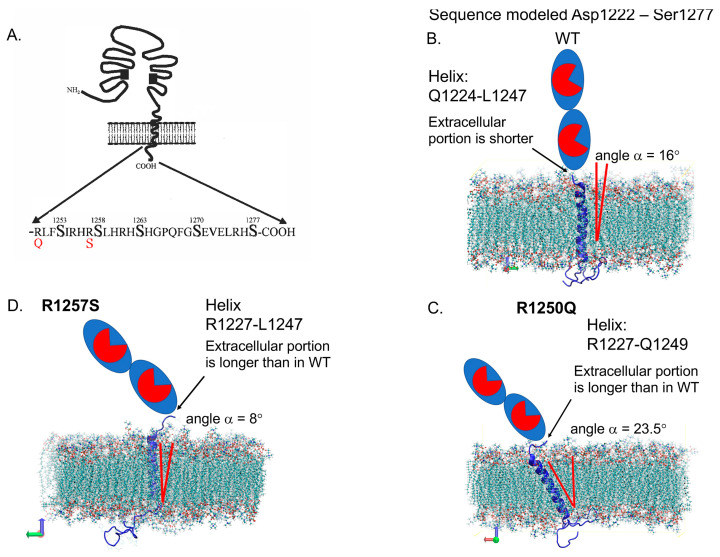
Localization of AD-associated *ACE* mutations on the cytoplasmic tail. (**A**) Schema showing the localization of two mutations in the cytoplasmic tail of ACE (R1250Q and R1257S)-adapted from [65]. (**B**–**D**). MD simulations of the transmembrane and cytoplasmic domains of ACE in POPC–lipid membrane. Sequence modelled from Asp1222 to Ser1277. Shown for WT (**B**) is a helix span from Q1224 to L1247, for R1250Q ACE mutant (**C**) helix span from R1227 to Q1249, and for R1257S ACE mutant (**D**) helix span from R1227 to L1247. The average angles of transmembrane helices with the lipid bilayer in mutant ACEs were changed in comparison with WT ACE.

In WT ACE, the average angle between the transmembrane helix, which in this simulation spans from Gln 1224 to Leu 1247, and the Z-axis of the lipid bilayer is 16 degrees (Figure 4B). In the R1250Q mutant, the transmembrane helix shifts toward the extracellular space and spans from Arg 1227 to Gln 1249, and its angle with the membrane bilayer is 23.5 degrees. (Figure 4C). Unlike in WT ACE, the neutral, albeit still polar, glutamine sidechain of Q1250 is much less effective at interacting with the negatively charged phosphate groups, allowing the transmembrane helix to tilt until it reaches the next positively charged sidechain at R1255. In the R1257S mutant, the transmembrane helix is formed between R1227 and L1247, whereas its average angle with the lipid bilayer is 8 degrees. (Figure 4D). The rationale for this decrease in the angle in R1257S mutant is not immediately obvious. 

Visual analysis of the MD trajectory suggests that the charged residues in the cytoplasmic portion of WT ACE form coulombic interactions with the phosphate groups of POPC. This is expected because the cytoplasmic portion of WT ACE contains five arginine residues, R1250, R1255, R1257, R1261, and R1275, and only two glutamic acid residues, E1271 and E1273. Distribution of the proximal positive charges of the arginine residues on the inner surface of the membrane is likely to create a lateral tension, leading to a 16-degree tilt in the orientation of the helix. 

In the R1257S mutant, the transmembrane helix remains anchored at R1250 on the inner part of the lipid bilayer, similarly to what occurs in WT ACE. We hypothesized that because the number of positive charges in the R1257S (and R1250Q) mutant is reduced by one (out of five positive and two negative residues), the requirements for membrane association of the charged residues in the cytoplasmic portion may be relaxed, making the interactions between the transmembrane domain of ACE and the lipid bilayer more dominant relative to those of the cytoplasmic domain. To investigate this possibility, we analyzed the distances between Q1225, located in the extracellular portion of ACE, and the amino acids with charged sidechains (R1250, R1255, R1257, R1261, E1271, E1273, R1275) in the cytoplasmic region of ACE. We observed that during MD simulation, these distances in both the mutants are greater, and the overall distribution is less focused, suggesting that their effect on the transmembrane helix orientation is indeed diminished. A representative example is shown in Appendix A for the distances between Q1225 and R1275 in WT ACE and both mutants.

These findings suggest that there is crosstalk between the extracellular and cytoplasmic portions of ACE. Even for mutations located in the cytoplasmic portion of ACE, the geometry of the extracellular N- and C-domains may be affected with respect to (1) the distance from the membrane, (2) the orientation of the domains relative to the membrane, and (3) the degree of dimerization. Hence, despite their cytoplasmic location, these mutations are likely to have direct effects on the conformation of ACE at the membrane, and possibly on the extent of ACE dimerization Thus, they are predicted to alter ACE catalytic activity and substrate specificity, especially involving long substrates such as Aβ42.

Increased female susceptibility to Alzheimer’s disease was recently reported in the carriers of one of the ACE mutations in the cytoplasmic tail—R1250Q (rs4980). Twelve of the thirteen AD patients with this mutation were women [42]. We have detected significant differences in the conformation of urinary ACE in men compared to women, which may be caused by differential glycosylation (specifically, sialylation) of kidney ACE (i.e., the source of ACE in urine) [66]. Protein glycosylation is involved in proper protein folding, protein quality control, transport of proteins to specific organelles, and sensitivity to shedding [8], including for ACE, which has 17 potential glycosylation sites. The sex-specific differences in tissue ACE glycosylation that we have identified [66] may be associated with differential disease susceptibility. One example is provided by structural differences between male and female ACE in certain neurons. Differences in ACE glycosylation in brain (striatal ACE) and lung ACE may be responsible for significant differences in substrate specificity of some brain-specific ACE substrates (i.e., substance P and substance K), at least in rats [67]. Therefore, it is logical to suggest that the R1250Q mutation may significantly impair Aβ42 cleavage in female carriers of this ACE mutation but not exhibit a similar harmful effect in males. 

In addition to the increased female prevalence in AD patients with the R1250Q ACE mutation, the lifetime risk for AD is nearly two-fold greater in women than in men [68]. Therefore, we speculate that gender-specific differences in ACE glycosylation, such as we have identified [66], may occur in other LoF *ACE* mutations as well. An intriguing possibility raised by this hypothesis is that the increased prevalence of functional pain syndrome in women [69] may be caused in part by these gender differences in tissue ACE because one of the pain mediators, substance P, is a specific substrate for brain ACE [67]. 

### 3.5. Conformational Fingerprinting of ACE in Human Brain

We previously demonstrated that the mAb binding pattern to ACE derived from different organs is tissue-specific and determined by alterations in ACE glycosylation/sialylation that differently occur in different organs/tissues, i.e., the concept of ACE conformational fingerprinting [70]. To characterize brain ACE, we compared the “conformational fingerprint”-binding patterns of 22 mAbs to different epitopes on the ACE N- and C-domain [30] with that for our “gold standard” lung ACE, derived from endothelial cells [70], and with kidney ACE, which originates from proximal tubule epithelial cells (Figure 5). Substantial differences in the conformational fingerprints of ACE from brain homogenates compared to lung homogenates (Figure 5A) may be attributed to differential ACE glycosylation in the endothelial cells of the lung and in ACE-positive cells in the frontal cortex of the brain (endothelial cells and neurons) [9,10]. Specific glycosylation sites in the ACE protein include Asn82 (localized in the epitopes for mAbs 3A5/3G8/5B3), Asn45 (mAbs 6C8/6H6/2D1) and perhaps Asn131 (epitope for mAb 2D7), Asn685 (epitope for mAbs 1E10/3C10/4E3/2H9), perhaps Asn913 (possibly in the epitope for mAb 8H1), and Asn1196 (epitope for mAb 4C12); see epitope mapping previously described in [30].

A surprising observation is that the conformational fingerprint of brain ACE is very similar (although not identical) to the conformational fingerprint of kidney ACE (Figure 5B). This similarity contrasts with previously described differences in mAb binding patterns (i.e., likely glycosylation pattern) between ACE from other epithelial cells, specifically from prostate and from adrenal glands, which have embryonic origin from the neural crest (Figure 3A,B in [70]). Theoretically, differences in mAb binding patterns in brain ACE compared to lung ACE may be due to differential ACE glycosylation in the brain endothelial cells and cortical neurons (the source of ACE in the brain) compared to the lung. Moreover, brain ACE demonstrated an additional, lower MW ACE band [67,71] with an alternative glycosylation pattern [71].

Additional support for the hypothesis that glycosylation (and perhaps more specifically, sialylation) may determine the observed differences in mAb binding to brain (and kidney) ACE versus lung ACE (Figure 5) is derived from calculation of the 2H9/2D1 binding ratio. We recently determined that this ratio is a marker of the extent of ACE sialylation at certain glycosylation sites in ACE. Binding of mAbs with Asn45 or Asn117 in their epitopes was dramatically decreased if these glycosylation sites were sialylated [66]. This prior study demonstrated that the 2H9/2D1 binding ratio for female urinary ACE (~10) was four-fold higher in comparison to male urinary ACE, while the 2H9/2D1 binding ratio for female and male lung tissues were similar and relatively low (about 3.0). In the present study, we observed that the 2H9/2D1 binding ratio for human male brain and kidney ACEs were extremely low at approximately four-fold less than for human male lung ACE (Figure 5, insert). This observation matches well with the established fact that lung ACE contains ~six-fold more sialic acids than kidney ACE [72]. Therefore, we conclude that the similarity in conformational fingerprints for brain and kidney ACEs (Figure 5) are attributed primarily to differences in ACE sialylation.

## 4. Conclusions

Amyloid peptide Aβ42 is a primary constituent of amyloid plaques in the brain and is cleaved by the N-domain active center of ACE [4]. Therefore, a decrease in tissue ACE activity has been proposed as a possible risk factor for Alzheimer’s disease (AD). Prior work has identified several ACE mutations as significantly associated with AD development. In the current study, we analyzed 31 available SNP databases for loss-of-function *ACE* mutations using PolyPhen-2 scores (Appendix A) and our extensive knowledge of ACE biology. Because approximately 4–5% of the general population are carriers of different LoF *ACE* mutations, we hypothesized that subjects with heterozygous LoF ACE mutations are at risk for the development of Alzheimer’s disease, especially for those LoF mutations that result in reduced ACE activity. 

Approximately 20–30% of these LoF *ACE* mutations are transport-deficient ACE mutations. The most exciting and potentially impactful consequence of our analysis is that it suggests a potential new therapeutic approach for patients with Alzheimer’s disease. Those who have transport-deficient ACE mutations theoretically may be treated with FDA-approved medications (chemical and pharmacological chaperones and proteosome inhibitors), either as a preventive intervention for high-risk patients or as a therapeutic approach for those with clinical disease. Therefore, we believe that an important future step will be clinical trials of these agents in patients who are identified as carriers of transport-deficient *ACE* mutations.

## Figures and Tables

**Figure 5 biomedicines-12-00162-f005:**
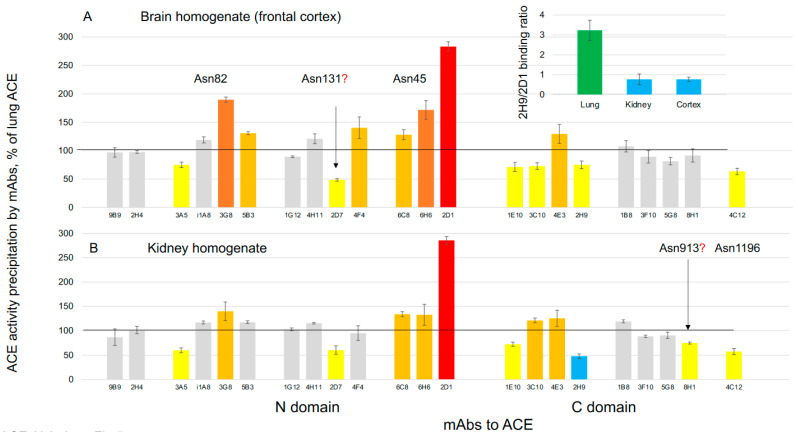
Conformational mAb fingerprinting of brain ACE. ACE activity was precipitated from human cortex homogenate (**A**), human kidney homogenates (**B**), and from human lung homogenate. Lung samples were used as a control (100%) for the comparison of 22 mAbs to different epitopes on the N- and C-domains of human ACE. Values of ACE activity precipitation with different mAbs characterizing ACE conformation were means ± SD from at least 2 independent experiments with duplicates or triplicates in each experiment. Data were expressed as a % of ACE activity from cortex and kidney homogenates precipitated by different mAbs from that for human lung. Orange bars—increase of ACE precipitation more than 20%, brown bars—more than 50%, red—more than 2-fold. Yellow bars—decrease of ACE precipitation more than 20%, blue bars—decrease more than 50%.

**Table 1 biomedicines-12-00162-t001:** Analysis of missense mutations in the *ACE* gene using in silico tools.

Prediction (Proportion/Count)	PolyPhen-2	SIFT4G	VEST4	REVEL	Majority Vote	Sum of MAFs *(%)	Mean/SEM of MAFs ** (%)
Damaging	0.34/416	0.46/573	0.54/663	0.22/268	0.33/412	0.81	0.0031/0.0005
Possibly Damaging	0.16/203	-	-	-	-	0.46	0.0037/0.0008
Benign/Tolerant	0.46/570	0.50/616	0.46/576	0.74/921	0.54/668	1.19	0.0031/0.0004
n/a	0.04/50	0.04/50	0/0	0.04/50	0.13/159	0.27	0.0028/0.0008

Variants in ACE genes (Appendix A) were mapped to the ACE protein sequence (UniProt accession: P12821) and functional predictions for all mapped nsSNVs were made using four different in silico tools (see Section 2). Note: Sum (*) and mean (**) of minor allele frequencies (MAFs) obtained from gnomAD v2.1.1 exome dataset. Common alleles with AF ≥ 0.5% were excluded (see Section 2; Appendix A).

**Table 2 biomedicines-12-00162-t002:** *ACE* mutations associated with different diseases, including Alzheimer’s disease.

#	Genetic Position	Amino Acid Position (Mature Protein)	Polymorphism and/or (Reference)	PolyPhen-2 Score (HVAR)	Minor Allele Frequency (per 100,000)	Blood ACE, % of Mean
1	p.Arg149Leufs*53	R120Lfs	rs778759606; [37,44]	1.000	5.2	Low
2	p.Arg228Cys	R199C	rs141543325; [44]	1.000	**24**	
3	p.Tyr244Cys	Y215C	rs3730025; [44,45,47]	1.000	**1068**	45 [11] Low [48] *
4	p.Ala261Ser	A232S	rs4303; #	0.420	742	
6	p.Tyr266X	Y237X	rs121912704; [37,44]	1.000	2.5	Low
7	p.Gly267Arg	G238R	rs149412997; [44]	0.999	**33**	
8	p.Trp343X	W314X	rs200225958; [37,46]	1.000	0.8	Low
9	p.Pro351Leu	P322L	rs2229830; [44]	0.999	**24**	
10	p.Gly354 Arg	G325R	rs56394458; [44]	1.000	780	
11	p.Thr381Met	T352M	rs150466411; [44]	1.000	**85**	
12	p.Asp441fs	D412fs	[46]	1.000	0.4	Low
13	p.Arg482Cys	R453C	rs201540553; [44]	0.999	**19**	Low [48] *
14	p.Pro485Arg	P456R	rs28730839; [44]	**0.539**	**48**	
15	p.Pro505Ala	P476A	rs148943954; [44]	0.997	**59**	
16	p.Arg561Gln	R532Q	rs4314; [44]	0.993	**78**	500 [49]
17	p.Asp592Gly	D563G	rs12709426; [42]	0.001	382	
18	p.His629Pro	H600P	rs201594771; [44]	0.003	506	
19	p.Ser660Cys	S631C	rs147429960; [44]	0.242	**93**	
20	p.Arg719Gln	R690Q	rs371010069; [24]	1.000	**24**	
21	p.Leu764Gln	L735Q	rs145819052; [44]	**0.869**	**25**	
22	p.Glu767Lys	E738K	rs148995315; [42,44]	0.989	**26**	
23	p.Iso798Val	I769V	rs117647476; [44]	0.002	213	
24	p.His861Tyr	H832Y	rs140056206; [44]	0.002	5.6	
25	p.Thr916Met	T887M	rs3730043; [42,44]	1.000	397	
26	p.Gly1013Ser	G984S	rs571848794; [44]	1.000	4.4	Low [48] *
27	p.Iso1018Thr	I989T	rs4976; [44]	1.000	**36**	
28	p.Leu1024fs	L995fs	[46]	1.000	0.4	Low
29	p.Asp1036Lys	N1007K	rs142947404; [42,46]	0.294	**80**	
30	p.Asp1058Tyrfs	D1029Yfs	[46]	1.000	0.4	Low
31	p.Arg1232His	R1203H	rs372282664; [44]	0.923	6.9	
32	p.Ser1238Pfs	S1209Pfs	[46]	1.000	0.4	Low
33	p.Arg1279Gln	R1250Q	rs4980; [42,44]	0.016	410	
34	p.Arg1284Cys	R1255C	rs375527470; [44]	0.978	5.4	
35	p.Arg1286Ser	R1257S	rs4364; [42]	0.001	733	
	**Combined frequency**		Probably damaging		2680	
			**Possibly** damaging		**73**	
			Benign		**3168**	
			All		**5921**	

ACE mutations, which names (based on mature ACE numbering) marked with red, were initially identified in relation to different diseases and then some of them were mentioned in association with Alzheimer’s disease (with different extent of association), and tested for ACE activity. Yellow highlights indicate association with AD. According to PolyPhen-2 scores [15], there are three possible predictions: (1) probably damaging, score ≥ 0.909; (2) **possibly ** damaging, **0.446 ** ≤ score ≤ **0.908**; (3) benign, score ≤ 0.445. Minor allele frequency values were marked according to frequency: bold **red**: >**1000**, red: >100; bold black: >**10**. Blood ACE *: This study (Appendix A), but calculated from values generated in [48]. #: LK Cuddy, 2020 personal observations.

## Data Availability

The datasets analyzed during the current study are available in the dbSNP repository [13]: (https://www.ncbi.nlm.nih.gov/snp/, accessed on 12 September 2022), and specifically, ACE mutations: (https://www.ncbi.nlm.nih.gov/snp/?term=ACE) were downloaded in tab-delimited text file format utilizing the “Send to:” link found on the dbSNP web search results page. Additional annotations used for filtering and validation of the variants were obtained from the UCSC Genome Browser database: https://hgdownload.soe.ucsc.edu/gbdb/hg38/snp/dbSnp155.bb.

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
