# Peer review of "Carriers of Heterozygous Loss-of-Function ACE Mutations Are at Risk for Alzheimer’s Disease"

_biomedicines, 2024, doi:10.3390/biomedicines12010162_

Round 1
Reviewer 1 Report
Comments and Suggestions for Authors
The manuscript entitled “Carriers of heterozygous loss-of-function ACE mutations are at risk for Alzheimer’s disease” hypothesized that subjects with heterozygous Loss-of-Function (LoF) ACE mutations are at risk for Alzheimer’s disease. This stems from the fact that ACE converts the neurotoxic Aβ42 to the antioxidant Aβ40 in the brain by the N domain active center of ACE. Hence, LoF mutations with lowered ACE expression/activity are expected to increase the brain levels of Aβ42 and amyloid aggregates. Therefore, a decrease in tissue ACE activity could be associated with Alzheimer’s disease pathogenesis. Herein, the authors analyzed SNP databases LoF mutations and compared them with the topology of known ACE mutations already associated with AD. Based on the analysis, the authors suggested several mechanisms by which ACE mutations may be associated with Alzheimer’s disease. The current review findings are interesting.
Comments:
1) To clarify for readers, the authors are advised to add the following statement to highlight the link between low ACE activity and Alzheimer’s disease: “ACE coverts Aβ42-to-Aβ40 in the brain by the N domain active center of ACE.4Therefore, a decrease in tissue ACE activity could be associated with Alzheimer’s disease pathogenesis”.
2) To avoid readers’ confusion, the authors are advised to explain the conclusion of the abstract (lines 27-29):
“Patients with transport-deficient ACE mutations (about 20 % of damaging ACE mutations) may benefit from preventive or therapeutic treatment with a combination of chemical and pharmacological (e.g., centrally acting ACE inhibitors)”.
This seems the opposite of what was described by the authors in lines 46-50: “Angiotensin-con- 46verting enzyme (ACE) is able to convert Aβ42-to-Aβ40 in the brain by the N domain active center of ACE.4 Therefore, a decrease in tissue ACE activity (either due to constitutive intrinsic mutations or ACE inhibitor treatment) could be associated with Alzheimer’s disease (AD) pathogenesis”. Please, clarify.
3) In the statistical analysis section, did the authors check data normality before proceeding to one-way ANOVA? Authors are advised to address this point and add the answers to the comment in the statistics section.
4) In the legend of Figure 5, the authors are advised to include information on the number of replicates from which data were extracted. I would suggest that the authors address this point and add the answers to the relevant figure legend.
5) In line 342, please clarify the statement “This study (Table S4), but calculated from values generated in39”.
6) The authors are advised to carefully revise the reference section. The authors are advised to unify the way they write the journal name. Sometimes it is written as a full name (reference # 26) while most references they are written as an abbreviation. Please, follow the journal instructions in this regard.
Comments on the Quality of English Language
Moderate editing of the English language is required.
Reviewer 2 Report
Comments and Suggestions for Authors
In the current paper, the authors hypothesize that heterozygous loss of function mutations in the enzyme Angiotensin-I converting enzyme (ACE) is a risk factor for the development of Alzheimer’s Disease (AD). The authors have analyzed the existing SNP databases and predicted that blood analysis of ACE levels might be an early indicator of AD. This paper is potentially interesting to the AD field. However, this paper is written poorly and in a highly disorganized way. The authors did not proofread the paper and this paper is full of grammatical and formatting errors. I recommend a major revision for this paper if not rejection. The authors should address the following concerns before resubmitting it again.
- The author should mention the full form of ACE in the abstract or the beginning, not at the end of the introduction.
- Section 3.1 of Results and Discussion should be a part of the introduction as the authors did not mention any results obtained from their analysis. They only mentioned what is already known.
- Table 1 has no table legends. There is only a note below the table. The rest of the tables also seem a bit disorganized e.g. Table S1 and S2 come after Table S3 and S4, though table legends for Table S1 and S2 come before S3 and S4, table S2 has “SP” in all rows of “Amino acid position” column. It is not very informative to have the same information in the entire column. The authors could have just put it in the table legends. Similarly, in another part of table S2 with the subtitle ”Indels or stop codons in mature ACE”, one whole column is empty. The authors should organize the tables in a more informative and less confusing way.
- Figure 4 has “ ACE. Alzheimer. Fig4” on the left corner of the figure, which seems a strange placement.
- In the method section, the catalog number is missing for all the reagents. Catalog number is important for the reproducibility of the method, so it is important to include it in the method section.
- Line 238-239: why the D allele that has increased ACE levels and caused increased lifespan is associated with the development of AD? Shouldn’t it be the opposite of what is stated in lines 238-239? Various places in the paper mentioned that low levels of ACE cause AD but why increased levels are associated with AD? I think the authors should reframe the sentence to convey the right message.
- Table 2 has bold and regular font in the same color to indicate certain mutations, which is very confusing and easy to miss. I would recommend using different colors for different mutations. The legends of the table mentioned, “Yellow highlights indicate association with AD”. However, there is no yellow highlight anywhere in the table. 12, 28,30,32 has a reference number 26 instead of polymorphism.
- The authors have mentioned that ACE has sex-associated differences associated with AD. Is the ACE gene X-linked?
- Figure 5 is represented in a very strange way, where the authors have color-coded the bars based on the fold differences in the % ACE activity. It is obvious to see the difference by looking at the bars. They should color code the bars if they are different genotypes/genes etc, not to point out the obvious fold change.
Comments on the Quality of English LanguageThe authors should have paid more attention to the formatting of the paper. This paper is full of formatting errors. The paper has all kinds of fonts, random bold font, spacing mistakes (e.g. lines 83, 106, 206,365, and incomplete and confusing sentences. All these errors make it very unpleasant to read. e.g. line 68-69, 293-294, 227-229 has a much bigger font compared to the rest of the text, random usage of bold font seems unnecessary and is annoying, lines 245-246, the sentence is incomplete, etc.
Round 2
Reviewer 1 Report
Comments and Suggestions for Authors
The authors have adequately addressed the raised comments. Thanks!
Reviewer 2 Report
Comments and Suggestions for Authors
Thanks for sharing the revised version. The authors have addressed the concerns raised by me and have explained the tables, which could not be changed and are automatically generated.